# Addressing shortfalls of laboratory HbA$_{1c}$ using a model that incorporates red cell lifespan

Yongjin Xu[1], Richard M Bergenstal[2], Timothy C Dunn[1], Ramzi A Ajjan[3]*

[1]Abbott Diabetes Care, Alameda, United States; [2]International Diabetes Center, Park Nicollet, HealthPartners, Minneapolis, United States; [3]Leeds Institute of Cardiovascular and Metabolic Medicine, University of Leeds, Leeds, United Kingdom

**Abstract** Laboratory HbA$_{1c}$ does not always predict diabetes complications and our aim was to establish a glycaemic measure that better reflects intracellular glucose exposure in organs susceptible to complications. Six months of continuous glucose monitoring data and concurrent laboratory HbA$_{1c}$ were evaluated from 51 type 1 diabetes (T1D) and 80 type 2 diabetes (T2D) patients. Red blood cell (RBC) lifespan was estimated using a kinetic model of glucose and HbA$_{1c}$, allowing the calculation of person-specific adjusted HbA$_{1c}$ (aHbA$_{1c}$). Median (IQR) RBC lifespan was 100 (86–102) and 100 (83–101) days in T1D and T2D, respectively. The median (IQR) absolute difference between aHbA$_{1c}$ and laboratory HbA$_{1c}$ was 3.9 (3.0–14.3) mmol/mol [0.4 (0.3–1.3%)] in T1D and 5.3 (4.1–22.5) mmol/mol [0.5 (0.4–2.0%)] in T2D. aHbA$_{1c}$ and laboratory HbA$_{1c}$ showed clinically relevant differences. This suggests that the widely used measurement of HbA$_{1c}$ can underestimate or overestimate diabetes complication risks, which may have future clinical implications.

*For correspondence:
R.Ajjan@leeds.ac.uk

## Introduction

High glucose exposure in specific organs (particularly eye, kidney, and nerve) is a critical factor for the development of diabetes complications (*Marcovecchio, 2017*; *Giacco and Brownlee, 2010*). Laboratory HbA$_{1c}$ is routinely used to assess glycaemic control, but studies report a disconnect between this glycaemic marker and diabetes complications in some individuals (*Cohen et al., 2003*; *Bonora and Tuomilehto, 2011*). The exact mechanisms for this are not always clear but, at least in some cases, likely related to inaccurate estimation of intracellular glucose exposure in the affected organs.

While raised intracellular glucose is responsible for diabetes complications (*Giacco and Brownlee, 2010*; *Brownlee, 2005*), extracellular hyperglycaemia selectively damages cells with limited ability to adjust cross-membrane glucose transport effectively (*Brownlee, 2005*). HbA$_{1c}$ has been used as a biomarker for diabetes-related intracellular hyperglycaemia for two main reasons. First, the glycation reaction occurs within red blood cells (RBCs) and therefore HbA$_{1c}$ is modulated by intracellular glucose level. Second, RBCs do not have the capacity to adjust glucose transporter GLUT1 levels and thus are unable to modify glucose uptake, behaving similarly to cells that are selectively damaged by extracellular hyperglycaemia (*Brownlee, 2005*). Therefore, under conditions of fixed RBC lifespan and glucose uptake, HbA$_{1c}$ mirrors intracellular glucose exposure in organs affected by diabetes complications. However, given the inter-individual variability in both glucose uptake and RBC lifespan (*Cohen et al., 2008*; *Khera et al., 2008*), laboratory HbA$_{1c}$ may not always reflect intracellular RBC glucose exposure. While variation in RBC glucose uptake is likely relevant to the risk of diabetes complications in susceptible organs, variation in red cell lifespan can affect haemoglobin glycation and HbA1c values, in turn compromising the accuracy of this glycaemic marker in predicting risk of complications. This explains the inability to clinically rely on laboratory HbA$_{1c}$ in those with haematological disorders

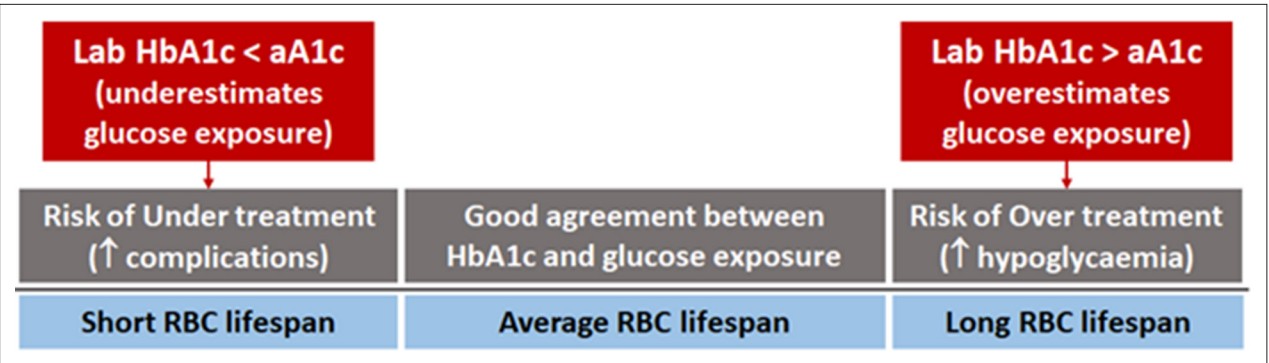

**Figure 1.** Individual red blood cell (RBC) lifespan can affect HbA$_{1c}$ and diabetes treatment. In some individuals, laboratory HbA$_{1c}$ can be misleading and resulting in undertreatment, thus increasing the risk of complications, or overtreatment, predisposing to hypoglycaemia.

characterised by abnormal RBC turnover (*American Diabetes Association, 2019*) and represents a possible explanation for the apparent 'disconnect' between laboratory HbA$_{1c}$ and development of complications in some individuals with diabetes (*Figure 1*).

A kinetic model, which considers individual variations in both RBC turnover and glucose uptake, has been developed to explain the disconcordance of the glucose-HbA$_{1c}$ relationship on individual level (*Xu et al., 2021a*). The current work aims to extend this model by providing a way to normalise against RBC lifespan variation when individual RBC lifespan becomes available. We propose a new clinical marker, which we term adjusted HbA$_{1c}$ (aHbA$_{1c}$), by adjusting laboratory HbA$_{1c}$ for a standard RBC lifespan of 106 days (*English and Lenters-Westra, 2018*) (equivalent to RBC turnover rate of 0.94 % per day). The new glyacemic marker, aHbA$_{1c}$, is likely to be the most accurate marker of organ exposure to hyperglycaemia and risk of future diabetes-related complications.

## Results

Of the 287 individuals in the original studies, 218 had predefined continuous glucose monitoring (CGM) coverage between at least two HbA$_{1c}$ measurements. Of these, 131 individuals had adequate continuous glucose data to estimate RBC lifespan and glucose uptake rate. The subject characteristics of this sub-cohort are presented in *Table 1*.

Mean (median, IQR) RBC lifespan was 94 (100, 86–102) days in those with T1D and 92 (100, 83–101) in those with T2D (*Figure 2*). In this cohort, the mean, median, IQR of the absolute difference between aHbA$_{1c}$ and laboratory HbA$_{1c}$ were 11.0, 3.9, 3.0–14.3 mmol/mol (1.0, 0.4, 0.3–1.3%) for T1D, and marginally higher at 15.1, 5.3, 4.1–22.5 mmol/mol (1.4, 0.5, 0.4–2.0%) for T2D subjects. As illustrated in the figure, those with the shorter RBC lifespan of 80 days showed around 22 mmol/mol (2%) lower laboratory HbA$_{1c}$ than aHbA$_{1c}$. This may lead to underestimating intracellular glucose exposure in susceptible organs, in turn increasing the risk of complications. In contrast, those with RBC lifespan of 130 days demonstrated higher laboratory HbA$_{1c}$ than aHbA$_{1c}$, which can give the impression of inadequate glycaemic control, leading to therapy escalation and predisposition to hypoglycaemia.

**Table 1.** Main characteristics of the cohort studied.

| N | 131 |
| --- | --- |
| Age [years; mean ± SD (range)] | 53.5 ± 13.7 (18, 77) |
| Gender, male [number (percentage)] | 86 (66%) |
| T1D [number (percentage)] | 51 (39%) |
| T2D [number (percentage)] | 80 (61%) |
| BMI [kg/m²; mean ± SD (range)] | 29.8 ± 5.9 (18.8, 54.1) |
| Duration of diabetes (years) | 17.7 ± 8.7 (2, 46) |
| Hypoglycaemic therapy | Multiple daily injections of insulin |

Data are presented as mean ± SD (min, max) or n (%)

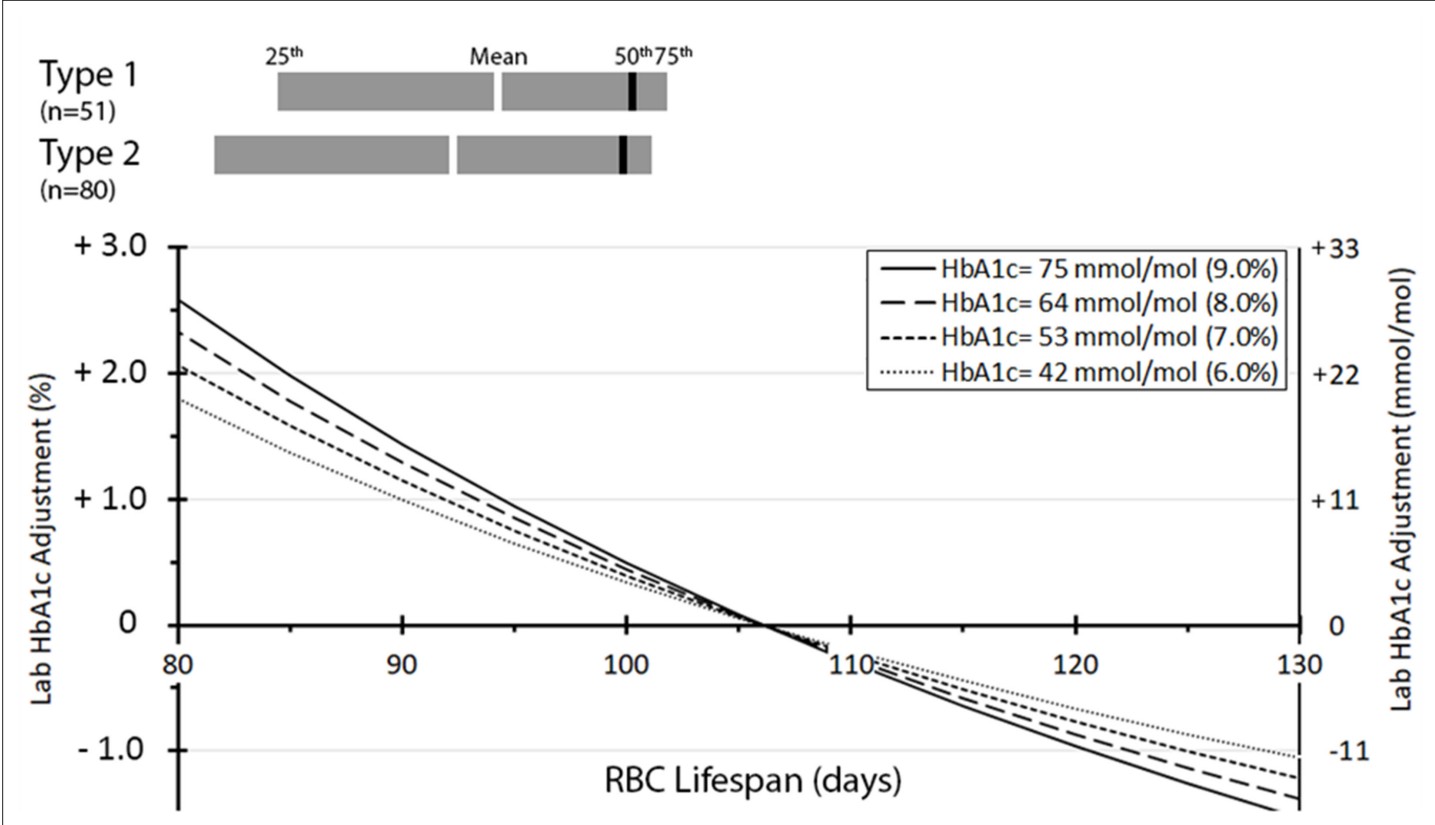

**Figure 2.** Distribution of red blood cell (RBC) lifespan for type 1 (n = 51) and type 2 (n = 80) diabetes and adjustment to laboratory HbA$_{1c}$ by RBC lifespan. The number (percentage) of individuals having HbA$_{1c}$ adjustments < 1 % (<11 mmol/mol), 1–2% (11–22 mmol/mol), 2–3% (22–33 mmol/mol), and >3% (>33 mmol/mol) were 90 (68%), 21 (16%), 12 (9%), and 8 (6%), respectively.

To further put these results into clinical context, two subjects with an identical laboratory HbA$_{1c}$ of 63 mmol/mol (7.9%) but different RBC lifespans of 89 and 107 days, would have RBC-lifespan-adjusted aHbA$_{1c}$ values of 78 mmol/mol (9.3%) and 62 mmol/mol (7.8%), respectively, indicating different future risk of diabetes complications. Another two individuals with different laboratory HbA$_{1c}$ of 60 mmol/mol (7.6%) and 75 mmol/mol (9.0%), and corresponding RBC lifespans of 89 and 107 days, would have identical aHbA$_{1c}$ value of 74 mmol/mol (8.9%). This would place them at similar risk of diabetes complications, despite the significantly different laboratory HbA$_{1c}$ values. Generally, in individuals with RBC lifespan of approximately 93–123 days, aHbA$_{1c}$ and laboratory HbA$_{1c}$ showed relatively small differences (<11 mmol/mol or 1 % when laboratory HbA$_{1c}$ < 64 mmol/mol or 8%). In this cohort, 90 (69%) subjects were within RBC lifespan range of 93–123 days, while 39 (30%) subjects had RBC lifespan below 93 days and 2 (1.5%) subjects above 123 days.

## Discussion

Variation in RBC lifespan and glucose uptake between individuals can lead to different laboratory HbA$_{1c}$ despite similar hyperglycaemic exposure in the organs affected by diabetes complications. In order to individualise care and assess the personal risk of hyperglycaemic complications, laboratory HbA$_{1c}$ levels should be adjusted to account for variability in RBC turnover through our proposed aHbA$_{1c}$. Without this adjustment, there is a risk of overestimating glucose levels that may cause hypoglycaemia through the unnecessary escalation of diabetes therapies, or alternatively, underestimation that may lead to undertreatment and subsequent high risk of complications. In addition, there are implications for the diagnosis of prediabetes and diabetes, as there may be misclassifications if the diagnosis is based solely on laboratory HbA$_{1c}$ levels due to variable RBC lifespan across individuals.

RBC removal by senescence and erythrocyte apoptosis are complex processes, which can be affected by the presence of hyperglycaemia and known to vary both within and across individuals

(*Lang et al., 2012*). In the meantime, potential differences in RBC glucose uptake (*Khera et al., 2008*) can also affect the relationship between blood glucose and HbA$_{1c}$. Several mathematical models (*Malka et al., 2016*; *Fabris et al., 2020*) have been developed to estimate laboratory HbA$_{1c}$ from glucose levels or time in range, emphasising the importance of this area. Accurate estimation of 'clinically relevant HbA1c' will allow each person with diabetes to have an individualised glycaemic target that ensures adequate treatment, thus reducing the risk of complications while minimising hypoglycaemic risk.

A unique feature of our model (*Xu et al., 2021a*) is the inclusion of individual-specific RBC lifespan and glycation rate in the calculations. A weakness of this model, however, is the absence of a direct measure of RBC lifespan, which remains an estimate based on a mathematical calculation. However, the ability of the model to reflect laboratory HbA$_{1c}$, as we have previously shown, indicates a good level of accuracy at estimating RBC lifespan ( *Xu et al., 2021c*). In addition, the method is far simpler than complex methods for estimating RBC lifespan through labelling experiments that are not suited for routine clinical practice (*Cohen et al., 2008*). Future work may determine whether other measures, such as reticulocyte count or red cell distribution width (*Brodksy, 2021*; *Kameyama et al., 2018*; *Kameyama et al., 2020*), can further be added to the model to further improve the accuracy of estimating RBC lifespan and this remains an area for future research.

Since aHbA$_{1c}$ reflects intracellular glucose exposure in RBCs, it is difficult to directly compare with extracellular glucose-derived glycaemic markers such as average glucose or time in range. As an intracellular marker, aHbA$_{1c}$ should correlate with intracellular glucose levels, therefore providing a potentially accurate measure of glucose exposure of organs susceptible to diabetes complications. We summarise the advantages and drawbacks of different methods that measure average glucose control in *Appendix 1—table 2*.

Importantly, our study demonstrates that laboratory HbA$_{1c}$ does not necessarily reflect intracellular glucose exposure of organs prone to diabetes complications. However, future work is required to show that adjusted A$_{1c}$ is a better predictor of diabetes complications than laboratory HbA$_{1c}$. Moreover, it is unclear whether the use of aHbA$_{1c}$ reduces the risk of hypoglycaemic complications as compared to reliance on laboratory HbA$_{1c}$, and these remain areas for future research.

In conclusion, quantitative aHbA$_{1c}$, derived from laboratory HbA$_{1c}$ and CGM readings, has the potential to more accurately assess glycaemic exposure of different organs, providing a safer and more effective glycaemic guide for the management of individuals with diabetes. Future testing in larger populations and different ethnic groups is required to further increase confidence in the model. This to be followed by large prospective clinical studies to test the relationship between aHbA$_{1c}$ and future microvascular/macrovascular diabetes complications as well as reducing the risk of hypoglycaemic exposure through avoidance of unnecessary therapy escalation.

## Materials and methods

CGM and laboratory HbA$_{1c}$ data from 139 type 1 (T1D) and 148 type 2 diabetes (T2D) patients, enrolled in two previous European clinical studies (*Bolinder et al., 2016*; *Haak et al., 2017*), were evaluated to calculate aHbA$_{1c}$ as detailed below. These studies were designed to evaluate the benefits of CGM in those with T1D and those with T2D using multiple daily injections of insulin. Both studies were conducted after appropriate ethical approval and participants gave written informed consent. A total of 6 months' CGM data were collected using the sensor-based flash glucose monitoring system (FreeStyle Libre; Abbott Diabetes Care, Witney, UK), while HbA$_{1c}$ was measured by a central laboratory (ICON Laboratories, Dublin, Ireland) at 0, 3, and 6 months of the study. For T1D participants, the mean age was 44 years (range 18–70 years), 17 (33%) of whom were females. For T2D, the mean age was 59 years (range 33–77 years), 28 (35%) of whom were females.

Each subject had at least one data section consisting of two HbA$_{1c}$ measurements connected by CGM data. Since the kinetic parameters are more sensitive to the data sections with larger between-day glucose changes, the parameters were successfully estimated for those individuals with sufficient day-to-day glucose variability, as evidenced by the model fit of RBC life converging between 50 and 180 days. These individual RBC lifespans or turnover rates were calculated according to previous model (*Xu et al., 2021a*) that considers both RBC turnover rate and glucose uptake. Briefly, the model aligns laboratory HbA$_{1c}$ and the contemporaneous CGM-derived estimate of HbA$_{1c}$ under optimal values for RBC turnover and glucose uptake of each individual. Since there is no simple clinical assay

for RBC turnover and glucose uptake, these RBC parameters are estimated using a numerical method such that differences between laboratory HbA$_{1c}$ and CGM-derived estimate are minimized. While the parameter identification method can be performed by repeated permutations across all reasonably possible values for RBC lifespan and uptake, our approach uses a far more efficient and reliable numerical method, as previously described (*Xu et al., 2021a*). Detailed model description and derivation are provided in Appendix 1. Deriving from the same model, we constructed aHbA$_{1c}$ (Equation 1) that adjusts laboratory HbA$_{1c}$ for individual RBC turnover variation for potential clinical use.

$$aHbA1c = \frac{HbA1c}{HbA1c + \frac{k_{age}^{ref}}{k_{age}}\left(1 - HbA1c\right)}$$

(1)

In an approximation, $aHbA1c \approx \frac{k_{age}}{k_{age}^{ref}} HbA1c$, where $HbA_{1c}$ is laboratory HbA$_{1c}$, $k_{age}$ is individual RBC turnover rate (%/day), $k_{age}^{ref}$ is standard RBC turnover rate (0.94%/day). HbA$_{1c}$ and aHbA$_{1c}$ are in NGSP unit and decimal values should be used. For example, 8 % HbA$_{1c}$ should be applied as 0.08. Equation 1 for IFCC unit is available in Appendix 1.

Under the assumption of individually constant RBC life, the relationship between RBC turnover rate ($k_{age}$), RBC lifespan ($L_{RBC}$) and mean RBC age ($MA_{RBC}$) can be inter-converted using the simple formula: $2 * MA_{RBC} = L_{RBC} = \frac{1}{k_{age}}$. Therefore, 0.94%/day standard RBC turnover rate is equivalent to 106 days of RBC life and 53 days of mean RBC age. Of note, the adjustment is not linear, decreasing RBC lifespan corresponds to more pronounced aHbA$_{1c}$ adjustment than a seemingly comparable increase in RBC lifespan. All calculations in this study were done with Python/SciPy (*Virtanen et al., 2020*) software package.

Full derivation of the model is further provided in Appendix 1.

## Acknowledgements

This work was funded by Abbott Diabetes Care.

## Additional information

### Competing interests

Yongjin Xu: YX is an employee of Abbott Diabetes Care. Richard M Bergenstal: RMB has received research support, has acted as a consultant, or has been on the scientific advisory board for Abbott Diabetes Care, Ascensia, DexCom, Eli Lilly, Hygieia, Johnson & Johnson, Medtronic, Merck, Novo Nordisk, Onduo, Roche, Sanofi and United Healthcare. RMB's employer, non-profit HealthPartners Institute, contracts for his services and no personal income goes to RMB.. Timothy C Dunn: TCD is an employee of Abbott Diabetes Care. Ramzi A Ajjan: RAA received no payment for this work but has had research support and/or Honoraria from Abbott Diabetes Care, NovoNordisk, Eli Lilly, Johnson & Johnson, Boehringer Ingelheim, Bayer, Sanofi and AstraZeneca..

### Funding

| Funder | Grant reference number | Author |
| --- | --- | --- |
| Abbott Diabetes Care | | Yongjin Xu<br>Timothy C Dunn |

The funders had no role in study design, data collection and interpretation, or the decision to submit the work for publication.

### Author contributions

Yongjin Xu, Conceptualization, Data curation, Formal analysis, Funding acquisition, Investigation, Methodology, Project administration, Software, Supervision, Validation, Visualization, Writing - original draft, Writing - review and editing; Richard M Bergenstal, Ramzi A Ajjan, Conceptualization, Formal analysis, Investigation, Methodology, Supervision, Validation, Visualization, Writing - original draft, Writing - review and editing; Timothy C Dunn, Conceptualization, Data curation, Formal analysis,

Funding acquisition, Investigation, Methodology, Resources, Supervision, Validation, Visualization, Writing - original draft, Writing - review and editing

### Author ORCIDs
Yongjin Xu http://orcid.org/0000-0001-9446-8402
Timothy C Dunn http://orcid.org/0000-0003-3487-2504
Ramzi A Ajjan http://orcid.org/0000-0002-1636-3725

### Ethics
Data were used from two previously published clinical studies in Europe (reference 10 and 11). Both studies were conducted after appropriate ethical approval and participants gave written informed consent.

### Decision letter and Author response
Decision letter https://doi.org/10.7554/eLife.69456.sa1
Author response https://doi.org/10.7554/eLife.69456.sa2

## Additional files

### Supplementary files
• Transparent reporting form

### Data availability
Data file for figures have been provided.

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

## Appendix 1

### Derivation of Equation (1): kinetic model review

The previously published kinetic model (*Xu et al., 2021a*) for glucose and HbA$_{1c}$ relationship led to Equation (1), shown in the main text. We further describe here the derivation of the model for the convenience of the reader. We also cover how the model can be used to estimate personal kinetic parameters for RBC glucose uptake and RBC turnover rate.

Our model assumes: (1) first-order dependencies for concentrations of both haemoglobin in RBCs and intracellular glucose; (2) newly generated RBCs have a negligible amount of glycated haemoglobin; (3) RBCs have a fixed life, so that they are generated constantly and eliminated from circulation when they reach an age that is individual-specific.

The rate of change in glycated and non-glycated haemoglobin in RBCs can be modelled by differential equations:

$$d\left[HbG\right]/dt = k_g\left[GI\right]\left[Hb\right] - r*\alpha*A1c \tag{a1}$$

$$d\left[Hb\right]/dt = k_{gen}/C - r*\left(1 - \alpha*A1c\right) - k_g\left[GI\right]\left[Hb\right] \tag{a2}$$

[*HbG*] and [*Hb*] are the concentrations of glycated and non-glycated haemoglobin, respectively, while [*GI*] is intracellular glucose concentration. The *kg* is the rate constant of haemoglobin glycation reaction in unit of (concentration*time)$^{-1}$, with a reported value of about 0.0019 dL/mg/day (*Higgins and Bunn, 1981*). *C* is the total haemoglobin concentration $C = [Hb] + [HbG]$. *A*1c is the fraction of glycated haemoglobin [*HbG*]/*C*, *r* is RBC removal rate in unit of concentration/time, $\alpha$ is a coefficient used to scale HbA$_{1c}$ to the fraction of glycated haemoglobin to be removed (has no units of measurement).

The glucose transporters on RBC membranes (GLUT1) follows Michaelis-Menten kinetic with a universal *KM* approximately 26 mM (*Ladyzynski et al., 2011*). Intracellular glucose can be modelled with $d\left[GI\right]/dt = V_{max}*\left[G\right]/\left(K_M + \left[G\right]\right) - k_c*\left[GI\right]$, where [*G*] is the extracellular glucose concentration and *kc* is the rate of glucose consumption within RBCs. The maximum rate *V*max should be proportional to the GLUT1 level on the membrane and we assume both $k_c$ and $V_{max}$ can vary between individuals. Since this process is fast, we use the equilibrium condition:

$$[G1] = \frac{V_{max}*[G]}{K_c*(K_M+[G])} = \frac{V_{max}}{K_M*k_c}g = \frac{k_{gly}}{K_g}g = \frac{k_{gly}}{K_g}\frac{K_M*[G]}{K_M+[G]} \tag{a3}$$

where $g = \left(K_M*\left[G\right]\right)/\left(K_M + \left[G\right]\right)$ and $k_{gly} = k_g*V_{max}/\left(k_c*K_M\right)$.

By definition, HbA$_{1c}$ is the fraction of glycated haemoglobin found in RBCs:

$$A1c = \left[HbG\right]/C = \left(C - \left[Hb\right]\right)/C.$$

In steady state, $d\left[Hb\right]/dt = d\left[HbG\right]/dt = 0$, Equation (a1) becomes

$$C*k_g/\left(\alpha*r\right) = \left[HbG\right]/\left(\left[GI\right]\left[Hb\right]\right).$$

Combining with Equation (a3):

$$\frac{C*k_g*V_{max}}{\alpha*r*K_M*k_c} = \frac{\left[HbG\right]}{g*\left[Hb\right]} \tag{a4}$$

By combining all parameters associated with cross-membrane glucose transport and glycation from the right-hand side of Equation (a4), we define the composite glycation rate constant $k_{gly} = k_g*V_{max}/\left(k_c*K_M\right)$, where *kg* and $K_M$ are universal constants for the non-enzymatic haemoglobin glycation reaction and glucose affinity to GLUT1, respectively. Therefore, *kgly* can vary between individuals depending on *kc* and *Vmax*.

We attribute the rest of the parameters to RBC turnover *kage* = $\alpha$*r/C, which leads to the definition of the apparent glycation parameter *K*:

$$K = k_{gly}/k_{age} = [HbG] / (g * [Hb])$$ (a5)

Under a hypothetical steady state of constant glucose level, $HbA_{1c}$ should reach an equilibrium level, which is the 'equilibrium $HbA_{1c}$' or $EA$. Since $C=[HbG]+[Hb]$, Equation (a5) can be re-written to $K = (C - [Hb]) / (g * [Hb])$. Applying the definition $HbA_{1c} = HbG/C = (C–[Hb])/C$, we have:

$$EA = \frac{g}{K^{-1} + g}$$ (a6)

This relationship approximates the average glucose and $HbA_{1c}$ for an individual with a stable day-to-day glucose profile.

From Equation (a3): $[GI] = \frac{V_{max}}{K_M * k_c} g = \frac{k_{gly}}{k_g} g$, and substituting into Equation (a6) gives:

$$[GI] = \frac{k_{age} * EA}{k_g * (1 - EA)}$$ (a7)

Imaging two individuals who have identical intracellular glucose level $[GI]_{k_{age}} = [GI]_{k_{age}^{ref}}$, one with $k_{age}$ and the other with reference $k_{age}^{ref}$. The relationship of their equilibrium $HbA_{1c}$ values is:

$$\frac{k_{age} * EA_{k_{age}}}{k_g * \left(1 - EA_{k_{age}}\right)} = \frac{k_{age}^{ref} * EA_{k_{age}^{ref}}}{k_g * \left(1 - EA_{k_{age}^{ref}}\right)}$$

Again, $kg$ is the universal composite rate constant for glucose haemoglobin composite reaction. Simplifying this equation we have:

$$EA_{k_{age}^{ref}} = \frac{EA_{k_{age}}}{EA_{k_{age}} + \frac{k_{age}^{ref}}{k_{age}} \left(1 - EA_{k_{age}}\right)}$$

In a steady state, $EA_{k_{age}}$ is the $HbA_{1c}$ under RBC turnover rate of $k_{age}$. The $EA_{k_{age}^{ref}}$ is the $HbA_{1c}$ under the same intracellular glucose level, when RBC turnover rate is a reference value of $k_{age}^{ref}$. Therefore, if we were to compare intracellular glucose exposure, steady-state $HbA_{1c}$ should be adjusted to a reference RBC turnover rate, which lead to the $aHbA_{1c}$ in Equation (a1) by replacing $EA_{k_{age}^{ref}}$ with $aHbA_{1c}$ and $EA_{k_{age}}$ with $HbA_{1c}$:

$$aHbA1C = \frac{HbA1C}{HbA1C + \frac{k_{age}^{ref}}{k_{age}} (1 - HbA1C)} = \frac{1}{1 + \frac{k_{age}^{ref}}{k_{age}} \left(\frac{1}{HbA1c} - 1\right)}$$ (1)

To simplify the above, an approximation to Equation (a1) is $aHbA1C \ \frac{k_{age}}{k_{age}^{ref}} \ HbA1c$. Note that $\frac{k_{age}}{k_{age}^{ref}} = \frac{M_{RBC}^{ref}}{M_{RBC}} = \frac{L_{RBC}^{ref}}{L_{RBC}}$, where $M_{RBC}$ is the mean RBC age and $L_{RBC}$ is the RBC lifespan.

The $HbA_{1c}$ and $aHbA_{1c}$ take NGSP unit in decimal form by default. For example, the decimal form of $HbA_{1c}$ of 8 % is 0.08. The unit for $k_{age}^{ref}$ and $k_{age}$ should be %/day.

When IFCC unit (mmol/mol) is used for $HbA_{1c}$ and $aHbA_{1c}$, Equation (a1) becomes:

$$aHbA1C = \frac{1092.9}{1 + \frac{k_{age}^{ref}}{k_{age}} \left(\frac{100}{0.0915 * HbA1c + 2.15} - 1\right)} - 23.5$$ (2)

## Estimations of $k_{gly}$ and $k_{age}$ from glucose and $HbA_{1c}$ data and prospective validation: kinetic model review

Our previous publication (*Xu et al., 2021a*) gave the following relationship by solving the differential Equation (a1):

$$A1c_t = EA + \left(A1c_0 - EA\right) \cdot e^{-\left(k_{gly} * g + k_{age}\right)t}$$ (a8)

Equation (a8) is suitable for a short time interval. For a longer time period, a recursive form is required:

$$A1c_z = EA_z \left(1 - D_z\right) + \sum_{i=1}^{z-1} \left[EA_i \left(1 - D_i\right) \prod_{j=i+1}^{z} D_j\right] + A1c_0 \prod_{j=1}^{z} D_j$$ (a9)

Equation (a9) describes HbA$_{1c}$ change from time 0 to time $z$. $A_{1c0}$ and $A_{1cz}$ are the starting and ending HbA$_{1c}$ values. The time period is split into $z$ time intervals with lengths of $ti$ ($i = 1,2,3, \ldots, z$), where $D_i = e^{-(k_{gly}*g_i+k_{age})t_i}$ and $EA_i = \frac{g_i}{K^{-1}+g_i}$ , and $g_i$ can be calculated from the average glucose ($AG$) in the time interval $g_i = (K_M * AG_i)/(K_M + AG_i)$.

The value $A_{1cz}$ is equivalent to calculated HbA$_{1c}$ (cHbA$_{1c}$) at the end of time interval $tz$. While shorter time intervals – such as 4–6 hr – are expected to produce better results, we have shown that a time interval of 24 hr has produced acceptable performance (**Xu et al., 2021a**). Equation (a9) is central to our kinetic model. To estimate personal parameters $k$gly and $k$age, one or more data sections are needed, where a data section contains two HbA$_{1c}$ measurements, one at the start of the time period and one at the end, with frequent (i.e. every 15 min) glucose levels in-between. The optimised individual $k$gly and $k$age pair should best align the HbA$_{1c}$ and cHbA$_{1c}$, minimising the preferred error function, such as mean difference or sum-squared difference.

Once an individual's $k$gly and $k$age pair are available, Equation (a9) is used to project future HbA$_{1c}$ if provided frequent glucose measurements. Therefore, prospective model validation is possible when multiple data sections are available, such that one or more can be held out of the parameter estimation to be used for prospective evaluation. **Appendix 1—table 1** summarises results (**Xu et al., 2021a**; **Xu et al., 2021c**; **Xu et al., 2021b**) when all but the last data section is used to determine the individual $k$gly and $k$age pairs, and the held-out final data section is used for evaluation. The agreement between the last HbA$_{1c}$ and cHbA$_{1c}$ is compared to the agreement between the last HbA$_{1c}$ and the glucose management indicator (GMI) (**Xu et al., 2021a**; **Xu et al., 2021c**; **Xu et al., 2021b**). These studies demonstrated the superior accuracy of the kinetic model compared to the existing GMI method.

**Appendix 1—table 1.** Summary of kinetic model validation studies.
The mean absolute deviation differences between calculated HbA$_{1c}$ (cHbA$_{1c}$) and glucose management indicator (GMI) are statically significant with $p < 0.0001$.

| Study | T1D SAP [22] | | DPV T1D [23] | | Replace/mpact [9] | |
|---|---|---|---|---|---|---|
| Country | Japan | | Germany | | Europe | |
| Subject count (male) | 51 (14) | | 352 (171) | | 120 (79) [TID 54 (37), T2D 66 (42)] | |
| Age median (range) | 42 (6–73) | | 12.5 (3–19) | | 52 (18–77) | |
| HbA$_{1c}$ test | Central lab | | POC+ central lab | | Central lab | |
| CGM device | Medtronic | | Abbott | | Abbott | |
| Method | cHbA$_{1c}$ | GMI (14-day AG) | cHbA$_{1c}$ | GMI (14-day AG) | cHbA$_{1c}$ | GMI (14- day AG) |
| Abs. dev. % (mmol/mol)  Mean | 0.11 (1.2) | 0.47 (5.1) | 0.32 (3.5) | 0.57 (6.2) | 0.31 (3.4) | 0.66 (7.2) |
| SD | 0.06 (0.7) | 0.46 (5.0) | 0.28 (3.0) | 0.55 (6.0) | 0.22 (2.4) | 0.46 (5.0) |
| Median | 0.10 (1.1) | 0.36 (3.9) | 0.26 (2.8) | 0.46 (5.0) | 0.27 (3.0) | 0.5 (5.5) |
| Average bias % (mmol/mol) | 0 (0) | –0.3 (–3.3) | 0 (0) | 0.4 (4.4) | 0 (0) | –0.6 (–6.6) |
| $R^2$ | 0.91 | 0.65 | 0.79 | 0.52 | 0.88 | 0.63 |

## Glycaemic marker comparisons

Given the importance of intracellular glucose level in diabetes management. We provide following table to compare the intracellular aspects of some frequently used glycaemic markers.

**Appendix 1—table 2.** Main characteristics of markers assessing average glycaemic control.

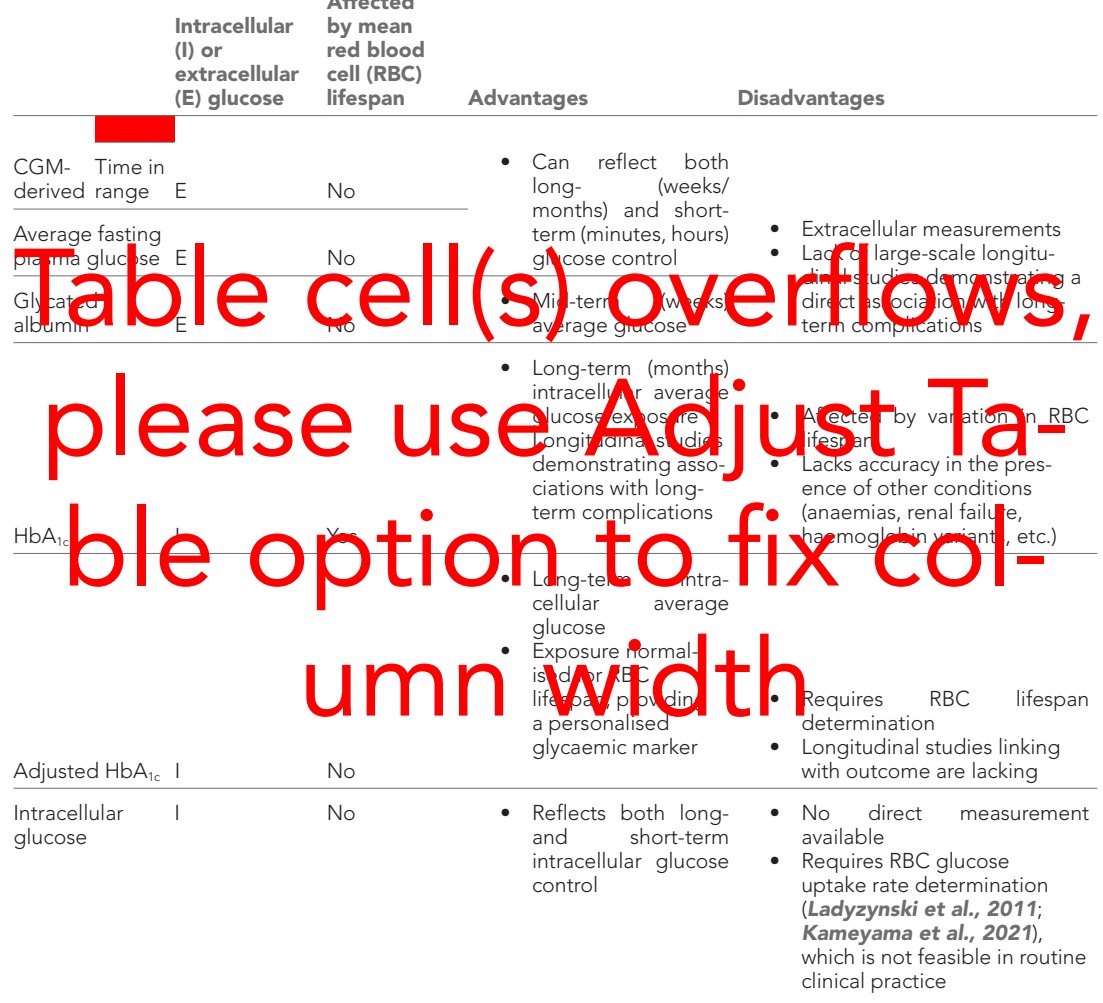

| | Intracellular (I) or extracellular (E) glucose | Affected by mean red blood cell (RBC) lifespan | Advantages | Disadvantages |
|---|---|---|---|---|
| CGM-derived Time in range | E | No | • Can reflect both long- (weeks/months) and short-term (minutes, hours) glucose control | |
| Average fasting plasma glucose | E | No | | • Extracellular measurements<br>• Lack of large-scale longitudinal studies demonstrating a direct association with long-term complications |
| Glycated albumin | E | No | • Mid-term (weeks) average glucose | |
| HbA$_{1c}$ | I | Yes | • Long-term (months) intracellular average glucose exposure<br>• Longitudinal studies demonstrating associations with long-term complications | • Affected by variation in RBC lifespan<br>• Lacks accuracy in the presence of other conditions (anaemias, renal failure, haemoglobin variants, etc.) |
| Adjusted HbA$_{1c}$ | I | No | • Long-term intracellular average glucose<br>• Exposure normalised for RBC lifespan, providing a personalised glycaemic marker | • Requires RBC lifespan determination<br>• Longitudinal studies linking with outcome are lacking |
| Intracellular glucose | I | No | • Reflects both long- and short-term intracellular glucose control | • No direct measurement available<br>• Requires RBC glucose uptake rate determination (*Ladyzynski et al., 2011*; *Kameyama et al., 2021*), which is not feasible in routine clinical practice |

