## [Decision Letter]

**Acceptance summary:**

This paper will be of interest to clinicians who provide care for persons with diabetes, educators who prepare these clinicians, as well as persons with diabetes who wish to be proactive participants in their own care. The calculation for an adjusted Hemoglobin A1c proposed by the authors can correct for individual red blood cell lifespan variations that can lead to misrepresentation of glycemic control. With the addition of a data-driven comparison to other means of assessing glycemic control, the adjusted HbA1c has the potential to improve care and subsequently decrease morbidity and mortality for persons with diabetes.

**Decision letter after peer review:**

Thank you for submitting your article "HbA_1c_ and Red Blood Cell Lifespan: Addressing shortfalls of the laboratory measure" for consideration by *eLife*. Your article has been reviewed by 3 peer reviewers, and the evaluation has been overseen by a Reviewing Editor and Nancy Carrasco as the Senior Editor. The following individuals involved in review of your submission have agreed to reveal their identity: Agnieszka Szadkowska (Reviewer #2); Masashi Kameyama (Reviewer #3).

Essential revisions:

1) A more complete description of the derivation of the model – more detail is required as to how the computations were done.

2) A comparison to other methods of assessing glycemic control (CGM, A1c). While assessment of how the authors' model predicts complications will not be required, a comparison (or at the very least, detailed discussion) of how it can compare to standard means of assessing glycemic control is necessary.*Reviewer #1:*

The strengths of this paper include that it builds on previous research and its sample size. However, the calculation for the individual RBC turnover rate, k age, is not included. The calculation for the adjusted HbA1c is too unwieldy for the clinician to use in practice where there is increased pressure to see many patients on a timely basis. If a "calculator" could be formulated allowing the clinician to plug in the relevant values and get the adjusted HbA1c, this could be widely used to improve patient care at the point of care which is the ultimate goal of this research.

The author's claims of a proposed adjusted HbA1c which is more accurate in predicting adverse outcomes due to diabetes is supported by the data.

There is a model to estimate RBC lifespan using reticulocyte count: Brodsky, R. A. Diagnosis of hemolytic anemia in the adult. In J. S. Timauer (Ed.) UpToDate, Retrieved February 17, 2020 from https://www.uptodate.com/contents/diagnosis-of-hemolytic-anemia-in-the-adult?search=hemolytic%20anemia&source=search_result&selectedTitle=1~150&usage_type=default&display_rank=1.*Reviewer #2:*

Laboratory HbA1c values are routinely used to assess glycemic control, but differences in red blood cell lifespan can affect hemoglobin glycation and HbA1c values. The authors developed a formula for adjusted HbA1c to account for red blood cell lifespan, which would better represent hemoglobin glycation and thus could better estimate the risk of disease complications.

The authors based their formula for aHbA1c calculation on laboratory measurements of HbA1c and CGM data performed with FreeStyle Libre. When using CGM systems, we often see falsified results for the duration of hypoglycemia, especially at night, due to sensor compression. The authors did not address these technical problems that may affect the aHbA1c result.

Using CGM we now have simple parameters to assess metabolic control of diabetes on the basis of CGM: TIR, GMI, CV. What I miss is a comparative analysis between these parameters and aHbA1c to convince readers that aHbA1c will be a better parameter for long term assessment for risk of complications.

The authors achieved their goal, and the results support their conclusions.

The use of aHbA1c in daily practice can be difficult because usually the clinician wants a ready result, but for research purposes, especially in patients with shorter red blood cell lifespans, it could be useful.

It will be interesting to see if its usefulness in estimating the risk of complications will be greater than TIR or GMI with CV.

Some comments on the methods and results:

Methods:

1. The study is based on CGM performed with FreeStyle Libre. From experience, we know that patients often have false hypoglycemia at night due to compression of the sensor. Has this been taken into account in the analysis?

2. Currently we use TIR and GMI to assess metabolic control in CGM users. Since aHbA1c, a new parameter will require additional calculations from me as a clinician, does it have advantages over these parameters in assessing the risk of chronic complications. It may be worth doing additional analysis comparing aHbA1c with these parameters.

3. As this is proof of concept, the details of creating the equation should be described, probably in some supplement. In addition, the idea behind the mathematical formula might help explain the equation to the readers. Secondly, the equation presented here is not clear in terms of units, and additionally operates slightly differently than the one in executed in provided excel file. The units for HbA1c here must be better emphasized. For example, if we assume HbA1c 9%, HbA1c in the numerator is treated as is (9%, % as a unit is preserved), but HbA1cs in the denominator have their units dropped (changed from 9% to 0.09). In parallel, the formula in excel does the same but multiplies the nominator by 100 (9% -> 900 %) and leaves HbA1cs in the denominator as is (and 1 becomes 100%). This creates some confusion and should be better explained, or an example calculation could be shown.

Results:

1. I think, that any characteristics of this group in needed. They come from already-reported cohorts but this is only a subgroup, so some basic clinical characteristics (if available) would be welcome.

2. Ad Figure 2 May be it should be consider adding a scatter plot showing individual patients or HbA1c measurements. Or, for simplicity, mark how many patients (N, %) had their HbA1c adjusted by up to 1, 1-2, 2-3 and >3%. This will provide a good estimate as to the range of applicability of your equation.

*Reviewer #3:*

The authors tried to adjust HbA1c to remove influence of erythrocyte lifespan. I admit the need for the adjustment, however, the study seems to lack the confirmation of their method.

1. The authors did not confirm the usefulness of their aHbA1c. The best way may be to confirm future diabetes complications as they mentioned, but it takes many years. I recommend to compare aHbA1c and average glucose derived by CGM.

2. The method requires complicated calculation from CGM data. I am not sure if the method is better than the simple average glucose derived CGM. Maybe, the kinetic method does not require steady state.

3. The value of k_gly_ was not stabilized; from 4.04E-6 to 9.95E-6. k_gly_ is a constant of non-enzymic process. It was estimated to be 6-10E-6, but the most recent one was 7.0E-6 (Kameyama et al. 2021). The average value of 5.86E-6 is smaller than the previously estimated value. I think that this instability is attributable to the nature of the calculation asking both k_gly_ and k_age_.

4. The value of HbA1c should be converted to IFCC value.

"While the National Glycohemoglobin Standardization Program (NGSP) is used to express HbA1c in many clinical research and medical care settings, NGSP is measured by an old standardized method and at the time of conception, HPLC was not able to distinguish true HbA1c from other products. HPLC technology later advanced, however the derived HbA1c value is adjusted to NGSP in the interest of consistency. IFCC provides a strict definition of iA1c as hemoglobin with a glycated valine in the N-terminal β-chain. Thus, iA1c value is preferred value for estimation of hemoglobin glycation." (Kameyama et al., 2021)

5. The authors chose random destruction model of erythrocytes, however "unlike some other species including mice, all normal human RBCs have about the same lifespan and thus exhibit non-random removal ( Franco, 2009 )." Kameyama et al., (2018) provided erythrocyte model based on Γ-distribution by Shrestha et al., (2016). Program Modification from random destruction model to the uniform distribution of RBC ages model (Γmodel requires 2 parameters, so model that every RBC dies at the same age would be better to program) may be troublesome, but I think it is worth.

6. Equation 1 needs the derivation. i.e. d HbA1c/dt = 0; -k_age_ HbA1c + k_gly_ AG(1-HbA1c) = 0, -k_age_^ref^ HbA1c + k_gly_ AG(1-HbA1c) = 0

Kameyama M, et al., Estimation of the hemoglobin glycation rate constant. Sci Rep. 2021 Jan 13;11(1):986. doi: 10.1038/s41598-020-80024-7.

Franco RS. The measurement and importance of red cell survival. Am J Hematol. 2009 Feb;84(2):109-14. doi: 10.1002/ajh.21298.

Kameyama M, Takeuchi S, Ishii S. Steady-state relationship between average glucose, HbA1c and RBC lifespan. J Theor Biol. 2018 Jun 14;447:111-117. doi: 10.1016/j.jtbi.2018.03.023.

Shrestha RP et al. Models for the red blood cell lifespan. J Pharmacokinet Pharmacodyn. 2016 Jun;43(3):259-74. doi: 10.1007/s10928-016-9470-4.

I think that the merit of the author's method is to obtain the erythrocyte lifespan. It would be interesting to compare mean erythrocyte age by Kameyama's equation (M_rbc_ = HbA1c / ((1-(2/3)HbA1c)k_g_ AG)) and k_age_.

---

## [Author Response]

Essential revisions:1) A more complete description of the derivation of the model – more detail is required as to how the computations were done.

As requested, we now provide further details on model derivation and validation. The additional details have been added to the main text and supplementary materials. More specifically, the following items have been added:

1. Derivation of the model and equation (1) (supplementary materials).

Please see under “Derivation of equation (1): kinetic model review”.

2. Kinetic parameter estimation (main text as below):

“Briefly, the model aligns laboratory HbA1c and the contemporaneous CGM-derived estimate of HbA1c under optimal values for RBC turnover and glucose uptake of each individual. Since there is no simple clinical assay for RBC turnover and glucose uptake, these RBC parameters are estimated using a numerical method such that differences between laboratory HbA1c and CGM-derived estimate are minimized. While parameter identification method can be performed by repeated permutations across all reasonably possible values for RBC lifespan and uptake, our approach uses a far more efficient and reliable numerical method, as previously described [9]. Detailed model description and derivation are provided in the supplementary materials.”

3. Using data from various cohorts, including different ages (paediatric, adult, old), populations (UK, DE, JP) and devices (Abbott, Medtronic), we further validate the model and compare with GMI (Table S1 added to supplementary material).

2) A comparison to other methods of assessing glycemic control (CGM, A1c). While assessment of how the authors' model predicts complications will not be required, a comparison (or at the very least, detailed discussion) of how it can compare to standard means of assessing glycemic control is necessary.

We have added to the Discussion and Supplementary material the differences, advantages and disadvantages of using different average glycemic measures for the management of individuals with diabetes with a summary table (Table S2).

Reviewer #1:The strengths of this paper include that it builds on previous research and its sample size. However, the calculation for the individual RBC turnover rate, k age , is not included. The calculation for the adjusted HbA1c is too unwieldy for the clinician to use in practice where there is increased pressure to see many patients on a timely basis. If a "calculator" could be formulated allowing the clinician to plug in the relevant values and get the adjusted HbA1c, this could be widely used to improve patient care at the point of care which is the ultimate goal of this research.

We would like to thank the reviewer for this pragmatic suggestion. Our model provides a method to calculate RBC lifespan from glucose and HbA_1c_ data. The optimization algorithm requires powerful software, inclusion of large data sets and complex calculations to accurately derive aHbA_1c_, hindering current use in routine clinical practice. However, it is envisaged that the software will be included in future CGM devices, which will be able to automatically calculate aHbA_1c_ once provided the necessary HbA_1c_ and CGM data.

To satisfy the reviewer, we do provide a simplified approach to calculate approximate aHbA_1c_ that can be provisionally used in clinical practice, provided there is access to high quality CGM data and two laboratory HbA_1c_ measurements. In following steps, the RBC turnover rate may be estimated from glucose and HbA_1c_ data in the supplementary materials, based on the previous publication [9].

1. For each data sections that contains a CGM trace and beginning and ending HbA_1c_ values, a series of daily average glucose are computed.

2. Feed the daily average glucose series to equation (s9) and perform a search for optimal parameter pair of k_gly_ and k_age_ that A1c_Z_ values best agree with lab values in all data sections available. Once k_age_ is available, equation (1) can be used to provide adjusted HbA_1c_. The RBC lifespan can be estimated roughly from reticulocyte count as pointed out by the reviewer:

RBC lifespan (days) ≈ 100 ÷ [Reticulocytes (percent) ÷ RLS (days)]

The reticulocyte life span (RLS) is 1.0, 1.5, 2.0, or 2.5 days at hematocrits of 45, 35, 25, and 15 percent, respectively. It can also be estimated by other measurements such as erythrocyte creatine, RDW.

The author's claims of a proposed adjusted HbA1c which is more accurate in predicting adverse outcomes due to diabetes is supported by the data.

Thank you for the comment, in the manuscript we state that this model has the potential to provide a more accurate measure of future diabetes complications but future large scale longitudinal studies are required to assess the clinical role of adjusted HbA_1c_.

There is a model to estimate RBC lifespan using reticulocyte count: Brodsky, R. A. Diagnosis of hemolytic anemia in the adult. In J. S. Timauer (Ed.) UpToDate, Retrieved February 17, 2020 from https://www.uptodate.com/contents/diagnosis-of-hemolytic-anemia-in-the-adult?search=hemolytic%20anemia&source=search_result&selectedTitle=1~150&usage_type=default&display_rank=1

We would like to thank the reviewer for this helpful suggestion. We are aware of the method estimating RBC lifespan using reticulocyte count but it has a number of drawbacks making it difficult to implement. First, it is a crude calculation of RBC lifespan and therefore the data generated are unlikely to be sensitive to small physiological changes in this parameter, encountered between different individuals. Second, it relies on steady state, indicating the need to take average of several measurements, adding difficulties to data collection and interpretation. Third, it requires repeated reticulocyte data, which are not usually performed in routine clinical practice.

We would have liked to analyse the role of incorporating reticulocytes into the model but we do not have the necessary data to undertake the work. We are evaluating multiple ways of estimating RBC lifespan and glucose uptake rate (added reference [20] in discussion) and will certainly consider adding reticulocyte count as an additional measure to investigate whether this improves the accuracy of the model.

Reviewer #2:Laboratory HbA1c values are routinely used to assess glycemic control, but differences in red blood cell lifespan can affect hemoglobin glycation and HbA1c values. The authors developed a formula for adjusted HbA1c to account for red blood cell lifespan, which would better represent hemoglobin glycation and thus could better estimate the risk of disease complications.The authors based their formula for aHbA1c calculation on laboratory measurements of HbA1c and CGM data performed with FreeStyle Libre. When using CGM systems, we often see falsified results for the duration of hypoglycemia, especially at night, due to sensor compression. The authors did not address these technical problems that may affect the aHbA1c result.

The results should be sensor independent as evidenced by a published validation using Medtronic sensor data [17]. The calculation depends on series of daily average glucose between HbA_1c_ measurements. The night-time low readings (due to compression or other artefacts) are usually short and should have a minimal effect on daily average glucose, particularly when large sets of data are collected.

In order to evaluate further, we divided CGM data into tertiles of hypoglycemic exposure (defined as glucose <54 mg/dl) during the period of 11PM – 6AM. The model validation was done by first calculating individual k_gly_ and k_age_ with the first data section and project HbA1c on the second held-out section. We validated the model by comparing the final lab HbA1c and CGM-derived HbA1c. The mean ± SD of the absolute deviation in CGM-derived HbA1c and lab HbA1c (NGSP %) in the lowest hypoglycemic tertile (0-1% or 0-4 mins/night) was 0.32±0.22, middle tertile (1-4% or 4-17 mins/night) was 0.32±0.20, and highest tertile (4-12% or 17-50 mins/night) was 0.30±0.23, showing no difference (p>0.1). Therefore, we conclude that nocturnal hypoglycemic events, whether true or artificial due to sensor compression, do not affect the accuracy of the kinetic model calculation.

Using CGM we now have simple parameters to assess metabolic control of diabetes on the basis of CGM: TIR, GMI, CV. What I miss is a comparative analysis between these parameters and aHbA1c to convince readers that aHbA1c will be a better parameter for long term assessment for risk of complications.

Thank you for raising this point; aHbA_1c_ is an intracellular marker of glycation and thus likely more relevant to diabetes complications that TIR or GMI, which are simply based on plasma glucose levels. We elaborated further in the discussion to clarify this point:

“As a intracellular marker, aHbA_1c_ should correlate with intracellular glucose levels, therefore providing a potentially accurate measure of glucose exposure of organs susceptible to diabetes complications”

and also added Table S2 comparing aHbA_1c_ with other average glycemic measures (please see above).

The authors achieved their goal, and the results support their conclusions.The use of aHbA1c in daily practice can be difficult because usually the clinician wants a ready result, but for research purposes, especially in patients with shorter red blood cell lifespans, it could be useful.

We would like to thank the reviewer for these positive comments.

It will be interesting to see if its usefulness in estimating the risk of complications will be greater than TIR or GMI with CV.

In theory, aHbA_1c_ should predict complication risk better than HbA_1c_, or other glycemic markers, by better reflecting intracellular glucose exposure in organs prone to diabetes complications. However, we fully agree with the reviewer that longitudinal studies are required to assess the potential superiority of aHbA_1c_ over other glycaemic markers and this will be part of our long-term strategy.

Some comments on the methods and results:Methods:1. The study is based on CGM performed with FreeStyle Libre. From experience, we know that patients often have false hypoglycemia at night due to compression of the sensor. Has this been taken into account in the analysis?

Please see our response above.

2. Currently we use TIR and GMI to assess metabolic control in CGM users. Since aHbA1c, a new parameter will require additional calculations from me as a clinician, does it have advantages over these parameters in assessing the risk of chronic complications. It may be worth doing additional analysis comparing aHbA1c with these parameters.

aHbA_1c_ should reflect intracellular glucose exposure in organs prone to diabetes complications and our data suggests it would be superior to HbA_1c_, which assesses RBC intracellular glucose exposure but fails to take into account potential artefacts generated by altered RBC lifespan. Also, aHbA_1c_ is potentially a better marker of future complication risk than average plasma extracellular glucose estimations such as TIR and GMI, which fail to take into account intracellular glucose exposure. However, large-scale longitudinal will be required to prove that aHbA_1c_ is a superior marker of diabetes complications and this is our long-term aim. We expanded the discussion to cover the above points and also added a Table addressing differences between aHbA_1c_ and GMI in relation to HbA_1c_ (please see response to comments above).

3. As this is proof of concept, the details of creating the equation should be described, probably in some supplement. In addition, the idea behind the mathematical formula might help explain the equation to the readers. Secondly, the equation presented here is not clear in terms of units, and additionally operates slightly differently than the one in executed in provided excel file. The units for HbA1c here must be better emphasized. For example, if we assume HbA1c 9%, HbA1c in the numerator is treated as is (9%, % as a unit is preserved), but HbA1cs in the denominator have their units dropped (changed from 9% to 0.09). In parallel, the formula in excel does the same but multiplies the nominator by 100 (9% -> 900 %) and leaves HbA1cs in the denominator as is (and 1 becomes 100%). This creates some confusion and should be better explained, or an example calculation could be shown.

We have added further clarifications as requested by the reviewer:

“HbA_1c_ and aHbA_1c_ are in NGSP unit and decimal values should be used. For example, 8% HbA1c should be applied as 0.08. Equation 1 for IFCC unit is available in the supplementary materials.”

Results:1. I think, that any characteristics of this group in needed. They come from already-reported cohorts but this is only a subgroup, so some basic clinical characteristics (if available) would be welcome.

Basic characteristics of the groups have been added as requested in Table 1.

2. Ad Figure 2 May be it should be consider adding a scatter plot showing individual patients or HbA1c measurements. Or, for simplicity, mark how many patients (N, %) had their HbA1c adjusted by up to 1, 1-2, 2-3 and >3%. This will provide a good estimate as to the range of applicability of your equation.

This has been done as requested:

“The number (percentage) of individuals having HbA1c adjustments < 1% (<11mmol/mol), 1-2% (11-22 mmol/mol), 2-3% (22-33 mmol/mol) and >3% (>33 mmol/mol) were 90 (68%), 21 (16%), 12 (9%), and 8 (6%), respectively.”

Reviewer #3:The authors tried to adjust HbA1c to remove influence of erythrocyte lifespan. I admit the need for the adjustment, however, the study seems to lack the confirmation of their method.1. The authors did not confirm the usefulness of their aHbA1c. The best way may be to confirm future diabetes complications as they mentioned, but it takes many years. I recommend to compare aHbA1c and average glucose derived by CGM.

Based on the Brownlee’s work [reference 5], it is the high intracellular glucose exposure that is the main cause of tissue damage and diabetes complications. However, average glucose levels do not necessarily reflect intracellular glucose exposure. Therefore, adjusted HbA_1c_, which reflects intracellular glycemia, is difficult to compare with average glucose (extracellular) and HbA_1c_ (affected by RBC lifespan). We added this to the discussion and also added Table S2 to summarise strengths and drawbacks of different average glycaemic measures.

2. The method requires complicated calculation from CGM data. I am not sure if the method is better than the simple average glucose derived CGM. Maybe, the kinetic method does not require steady state.

The kinetic parameter determination (for k_gly_ and k_age_) does not require a steady state. Please refer to the first answer in reviewer #1 for more details on determining kinetic parameters. The kinetic model relies on data from CGM-derived glucose levels and laboratory HbA_1c_. We demonstrated superior accuracy of the kinetic model compared with the average glucose derived HbA1c (GMI) in the supplementary material (Table S1). The parameter k_age_ (or RBC lifespan) need to be determined before adjusted HbA_1c_ can be calculated. Again, adjusted HbA_1c_ is an intracellular marker, while average glucose is not. This would be particularly relevant in individuals with apparent high laboratory HbA_1c_ and repeated hypoglycaemia or in those with low HbA_1c_ but high average glucose.

3. The value of k_gly_ was not stabilized; from 4.04E-6 to 9.95E-6. k_gly_ is a constant of non-enzymic process. It was estimated to be 6-10E-6, but the most recent one was 7.0E-6 (Kameyama et al., 2021). The average value of 5.86E-6 is smaller than the previously estimated value. I think that this instability is attributable to the nature of the calculation asking both k_gly_ and k_age_.

Although hemoglobin glycation is a non-enzymatic reaction with constant rate parameters, the reaction is limited by intracellular glucose levels rather than plasma glucose. According to Khera [7], RBC glucose uptake rate or intra/extracellular glucose concentration ratio varies person to person. For this reason, k_gly_, the overall glycation rate constant, should be an individual kinetic parameter, which explains various k_gly_ values reported in the literature. We highlight this in supplementary materials, summarizing our first publication of the model.

4. The value of HbA1c should be converted to IFCC value."While the National Glycohemoglobin Standardization Program (NGSP) is used to express HbA1c in many clinical research and medical care settings, NGSP is measured by an old standardized method and at the time of conception, HPLC was not able to distinguish true HbA1c from other products. HPLC technology later advanced, however the derived HbA1c value is adjusted to NGSP in the interest of consistency. IFCC provides a strict definition of iA1c as hemoglobin with a glycated valine in the N-terminal β-chain. Thus, iA1c value is preferred value for estimation of hemoglobin glycation." (Kameyama et al., 2021)

Thank you for raising this topic. We added equation (1) under NGSP unit system in the supplementary materials.

5. The authors chose random destruction model of erythrocytes, however "unlike some other species including mice, all normal human RBCs have about the same lifespan and thus exhibit non-random removal ( Franco, 2009 )." Kameyama et al., (2018) provided erythrocyte model based on Γ-distribution by Shrestha et al., (2016). Program Modification from random destruction model to the uniform distribution of RBC ages model (Γmodel requires 2 parameters, so model that every RBC dies at the same age would be better to program) may be troublesome, but I think it is worth.

Thank you for the helpful comment and for bringing new references to our attention.

The derivation starts with uniform age distribution or fixed-life model for RBC lifespan as suggested. We added the details to the supplementary materials. The additional complexity brought by glucose uptake following GLUT1 Michaelis-Menten kinetics and the approximations used has led to the steady-state equation (1). These approximations enable a robust application to real-world, non-steady state conditions. We have shown this reflects real world data well, given the accuracy of our validation studies [references 9, 17, 18]. We demonstrate below steady-state solution from a strict uniform age distribution, further confirming robustness of the model.

The steady-state equation (1) under the uniform age distribution is:

The circulating RBCs are produced within the last L days, where L is the expected lifespan. No RBC was lost until age>L. For a small fraction of RBC of age=t, where t<L:

hBgt= kg∗GI∗C∗ dtL∗t,where C is total hemoglobin (C = *Hb* + *HbG*)

Therefore, the total glycated hemoglobin is:

HbG= ∫OLkg∗GI∗ CL∗t∗dt= 12kg∗L∗GI∗C= kg2∗Kage∗GI∗CWhere,kage= 1L

Since HbA1c= HbGC, steady stateHbA1c: EA=kg2∗Kage∗GI

Equation (1) is about finding HbA1c under a reference k_age_ with the same intracellular glucose level. Therefore, the HbA1c with equivalent intracellular glucose under a reference Kage (GIkage= GIKageref) is:EAKageref= kagekageref∗ EAkage equation (1) becomes:equation (1f)AhBA1c= kagekageref∗HbA1c

This function is very close to equation (1) numerically. Note that equation (1f) is an approximation form of equation (1). In general, the differences between equation (1) and (1f) are less than 0.25% (NGSP) within +/- 20% change of RBC lifespan and HbA1c range of 5-10% (NGSP).

6. Equation 1 needs the derivation. i.e. d HbA1c/dt = 0; -k_age_ HbA1c + k_gly_ AG(1-HbA1c) = 0, -k_age_^ref^ HbA1c + k_gly_ AG(1-HbA1c) = 0Kameyama M, et al. Estimation of the hemoglobin glycation rate constant. Sci Rep. 2021 Jan 13;11(1):986. doi: 10.1038/s41598-020-80024-7.Franco RS. The measurement and importance of red cell survival. Am J Hematol. 2009 Feb;84(2):109-14. doi: 10.1002/ajh.21298.Kameyama M, Takeuchi S, Ishii S. Steady-state relationship between average glucose, HbA1c and RBC lifespan. J Theor Biol. 2018 Jun 14;447:111-117. doi: 10.1016/j.jtbi.2018.03.023.Shrestha RP et al. Models for the red blood cell lifespan. J Pharmacokinet Pharmacodyn. 2016 Jun;43(3):259-74. doi: 10.1007/s10928-016-9470-4.

Thank you for the new references, the majority of which have been incorporated into the current version of the manuscript. Please refer to our reply to essential revision (1) and additional supplementary materials for the derivation of equation 1.

I think that the merit of the author's method is to obtain the erythrocyte lifespan. It would be interesting to compare mean erythrocyte age by Kameyama's equation (M_rbc_ = HbA1c / ((1-(2/3)HbA1c)k_g_ AG)) and k_age_.

Future work is required for developing practical and robust methods that accurately measure RBC lifespan. Comparing to Kameyama's equation which has a universal glucose uptake (k_gly_), our model evaluates k_gly_ as individual-specific variable. Therefore, it is problematic to compare the two methods directly by k_age_ and M_RBC_. However, the ratio k_gly_/k_age_ may be comparable with M_RBC_ as per Kameyama’s work. While taking this caveat into account, we investigated the relationship between M_RBC_ and K and found a positive correlation (R=0.62, p<0.0001), providing additional evidence to support of our model.